# Potential CRISPR Base Editing Therapeutic Options in a Sorsby Fundus Dystrophy Patient

**DOI:** 10.3390/genes13112103

**Published:** 2022-11-12

**Authors:** Maram E. A. Abdalla Elsayed, Maria Kaukonen, Peter Kiraly, Jasmina Cehajic Kapetanovic, Robert E. MacLaren

**Affiliations:** 1Oxford Eye Hospital, Oxford University Hospitals NHS Foundation Trust, Oxford OX3 9DU, UK; 2Nuffield Department of Clinical Neuroscience, University of Oxford, Oxford OX3 9DU, UK

**Keywords:** TIMP3, age-related macular degeneration, macular choroidal neovascularisation, inherited retinal dystrophy, genetics, base editing, CRISPR

## Abstract

TIMP3 mutations are associated with early-onset macular choroidal neovascularisation for which no treatment currently exists. CRISPR base editing, with its ability to irreversibly correct point mutations by chemical modification of nucleobases at DNA level, may be a therapeutic option. We report a bioinformatic analysis of potential therapeutic options in a patient presenting with Sorsby fundus dystrophy. Genetic testing in a 35-year-old gentleman with bilateral macular choroidal neovascularisation revealed the patient to be heterozygous for a TIMP3 variant c.610A>T, p.(Ser204Cys). Using a glycosylase base editor (GBE), another DNA-edit could be introduced that would revert the variant back to wild-type on amino acid level. Alternatively, the mutated residue could be changed to another amino acid that would be better tolerated, and for that, an available ‘NG’-PAM site was found to be available for the SpCas9-based adenine base editor (ABE) that would introduce p.(Ser204Arg). In silico analyses predicted this variant to be non-pathogenic; however, a bystander edit, p.Ile205Thr, would be introduced. This case report highlights the importance of considering genetic testing in young patients with choroidal neovascularisation, particularly within the context of a strong family history of presumed wet age-related macular degeneration, and describes potential therapeutic options.

## 1. Introduction

Some mutations in TIMP3 are strongly associated with a rare inherited autosomal dominant macula disease which presents with early-onset macular choroidal neovascularisation (CNV). Sorsby fundus dystrophy (SFD) may result in bilateral loss of central vision due to either retinal pigment epithelium (RPE) atrophy or CNV leading to photoreceptor loss and irreversible blindness. There are currently 18 known TIMP3 mutations associated with SFD [1], with the majority occurring in the last exon of the TIMP3 gene (the C-terminus of the protein).

SFD is caused by mutations in the TIMP3 gene that are inherited in an autosomal dominant fashion. The protein is expressed and secreted by RPE and choroidal endothelial cells. Most of the SFD-associated mutations are located in the last exon of the TIMP3 gene [2]. They cause the gain or loss of a cysteine residues in the protein encoding the tissue inhibitor of metalloproteinases-3, resulting in aberrantly formed intermolecular di-sulphide bonds and the formation of TIMP3 dimers [3]. It has been demonstrated that SFD-associated TIMP3 variants are more resistant to turnover resulting in accumulation of TIMP3 at the level of Bruch’s membrane [4].

The TIMP3 protein has several functions in the retina including regulating Bruch’s membrane’s thickness, through the inhibition of matrix metalloproteinases (MMPs)—a group of zinc-binding endopeptidases that catalyse the degradation of the extracellular matrix (ECM) [5]. TIMP3 also inhibits MMPs and is capable of competitively inhibiting the binding of VEGF to its receptor VEGFR2. This has been shown to be mediated by the C-terminal domain, thereby inhibiting VEGF-mediated angiogenesis [6]. However, it has remained unclear whether SFD-associated TIMP3-variants maintain their ability to inhibit the binding of VEGF to VEGFR [7,8] and whether the variant proteins retain their other functions. It is a matter of controversy whether SFD-associated TIMP3-variants lead to a gain of function in which TIMP3 is more likely to accumulate, disrupting overlying RPE. Alternatively, the disease could develop via a loss of function in which the capacity for TIMP3 to inhibit MMP activity or VEGF signalling is reduced, leading to ECM dysregulation and neo-vascularisation [2].

There are currently no treatments for SFD. Gene replacement therapy has proven safe and efficient in treating recessive inherited retinal dystrophy, but is not suitable for dominant-negative diseases due to the need to silence the defective allele [9]. Recently developed CRISPR DNA base editing offers a tempting new approach to correct pathogenic single nucleotide variants, as it is able to correct, irreversibly, all transition variants [10] and even some transversion variants [11]. Unlike the traditional CRISPR-Cas9 gene editing [12], base editing does not introduce double-stranded DNA breaks, providing better in vivo safety. In addition, efficient base editing does not require cell division, which is important when targeting post-mitotic photoreceptors [13].

In principle, base editors consist of an inactivated CRISPR endonuclease (nickase) and a base editor -specific deaminase that is capable of introducing the DNA edits. Successful base editing requires a base editor-specific PAM site to be available adjacent to the targeted nucleotide in a position where it will place the variant within the construct’s editing window. Base editors are directed to the aimed target site with complementary guide RNAs. The two most commonly used base editors are the cytosine base editors (CBEs) and the adenine base editors (ABEs). CBE deaminates cytosine to uracil which is subsequently recognised by cell replication machinery as a thymine, resulting in a C-to-T transition [10]. In ABE-mediated DNA editing, the deoxyadenosine deaminase domain catalyses an adenine to inosine transition [12]. Inosine is recognised as guanine, and the original A:T base pair is replaced with a G:C base pair [12]. As base editors can target both DNA strands, all four transition variants (C→T, T→C, A→G, and G→A) can be corrected. The newest base editor subclass is the glycosylase base editors (GBEs) that are able to introduce C-to-G transversions [11]. The GBEs consist of a Cas9 nickase, a cytidine deaminase and an uracil-DNA glycosylase (Ung).

In this case report, we describe the clinical and genetic findings of a male Sorsby patient and explore current base editing options to treat his disease.

## 2. Materials and Methods

### 2.1. Clinical Examination

The patient underwent full consultative ophthalmic examination at the Oxford Eye Hospital. Clinical data and genetic testing were performed as part of routine clinical care and data and results were collected retrospectively. Retinal imaging methods included pseudo-colour fundus Optos images (Optomap P200; Optos plc, Dunfermline, UK) and spectral-domain optical coherence tomography (OCT) (Spectralis, Heidelberg Engineering, Inc., Heidelberg, Germany).

### 2.2. Genetic Testing

Informed consent for DNA blood sampling was taken from the patient and the sample was sent to the Oxford Regional Genetics Laboratories. A custom designed HaloPlex Target enrichment system (Agilent Technologies, Didcot, UK) was utilised to amplify the coding regions and intron/exon boundaries (+/− 10 bp) of the targeted genes and finally sequenced with Illumina MiSeq instrument (Illumina, San Diego, CA, USA) at the High-Throughput Genomics Group at the Wellcome Trust Centre for Human Genetics, Oxford. The utilised RetinalA_Macular panel includes the following genes: *ABCA4, BEST1, C1QTNF5, CDH3, CNGB3, EFEMP1, ELOVL4, FSCN2, GUCA1A, GUCA1B, IMPG1, IMPG2, PROM1, PRPH2, RP1L1, RS1,* and TIMP3. Detected variants were confirmed with Sanger sequencing. The genome annotation version GRCh37, hg19 was utilised in all analyses.

### 2.3. Base Editing

To explore possible base editing options, available PAM sites were searched and guide RNAs designed using the Benchling software (San Francisco, CA, USA) with construct-specific parameters (as detailed below) and the TIMP3 transcript ENST00000266085. The following base editing constructs, representing the most widely used base editors, were screened for available PAM sites: SpCas9-ABE8e [14], SaCas9-ABE8e [14], KKH-SaCas9-ABE8e [14], CasMINI-ABE8e [15], NG-SpCas9-ABE [16], SpCas9-CBE [17], SaCas9-CBE [17] and SpCas9-GBE [18]. Construct-specific PAM site requirements and editing windows are shown in Table 1.

Each designed guide was then analysed manually for possible bystander edits, which are undesirable edits that may occur if the editing window includes one or more additional identical bases to the aimed target base.

In silico analyses to assess the predicted effect of the introduced amino acid changes were performed using the published Sorting Intolerant from Tolerance (SIFT) [19], Polymorphism Phenotyping version 2 (Polyphen-2) [20] and Mutation Taster software [21].

## 3. Results

### 3.1. Clinical Results

A 35-year-old Caucasian gentleman presented with a two-month history of acute, painless vision deterioration in his right eye. He had previously been diagnosed elsewhere as having wet age-related macular degeneration. Apart from this, he had no significant ocular or medical history of note. There was no history of steroid use. He had a family history of age-related macular degeneration (AMD) diagnosed before the age of 60. The patient’s father, paternal aunt and paternal grandmother had presented in their 50s with acute reduction in vision.

Clinical examination revealed best corrected visual acuities of counting fingers and 6/4 in the right and left eyes, respectively. The pupillary responses were normal and there was absence of a relative afferent pupillary defect. Intraocular pressures were 15 mmHg OD and 18 mmHg OS and anterior segment examination was unremarkable. On fundus examination, the right macula exhibited greyish lesion at the fovea and intraretinal haemorrhage below the lesion. Left macula showed juxtafoveal temporal greyish lesion. There were no drusen in either eye (Figure 1A,B). OCT of the right macula showed intraretinal cysts and extensive subretinal hyperreflective material (Figure 1C). OCT of the left macula showed fibrovascular pigment epithelial detachment (PED) with a small accumulation of subretinal fluid. All the aforementioned morphological changes on multimodal imaging confirmed choroidal neovascularisation (CNV) in both eyes. Examination of the posterior segment was otherwise unremarkable. The patient was started on a course of monthly intravitreal injections of aflibercept 2 mg/0.05 mL to both eyes, with a good anatomical and functional response to treatment after the first four injections (Figure 2A–D). Monthly treatment was associated with good morphological response and no neovascular activity. After 15 months, the treatment was reintroduced with two-monthly aflibercept injections due to the slight neovascular reactivation in both eyes. In the first 3 years of treatment, the patient received 22 intravitreal injections of aflibercept in the right eye, and 20 injections in the left eye. Due to the waxing and waning of neovascular activity in both eyes, the patient was either on one-monthly or two-monthly intervals between the injections or on a pro re nata protocol. Thirty-one months after initial treatment, the patient maintained best corrected visual acuities of 5/60 and 6/6 in the right and left eye, respectively, and the intraocular pressures in both eyes have remained stable throughout the course of treatment which is ongoing.

### 3.2. Genetic Testing

Genetic testing revealed two heterozygous variants in the patient: a transversion variant in the TIMP3 gene (c.610A>T, p.(Ser204Cys)) and a deletion variant in the ABCA4 gene (c.4601del, p.(Leu1534fs)). The detected TIMP3 variant has previously been reported from individuals and families affected with SFD [22] and haplotype analysis has indicated it is a founder mutation in the British population [23]. We believe the *ABCA4* variant is an incidental finding due to the high carrier frequency in the general population of *ABCA4* variants. As a conclusion of the genetic testing, the TIMP3 variant was interpreted as the likely cause of the disease phenotype in the patient and possible treatment options were explored.

### 3.3. Base Editing

As an A-to-T transversion, the TIMP3 c.610A>T, p.(Ser204Cys) variant is not directly editable to the wild-type on DNA level with the currently available base editors. However, on amino acid level, the variant could be edited back to wild-type residue, serine, using a GBE: In wild-type sequence, the 204. serine is encoded by an AGC codon. In the c.610A>T, the codon is changed to TGC (cysteine). In theory, the TGC codon could be edited to a TCC if the reverse strand is targeted with a GBE, which would encode a serine residue (Option 1, Table 2). Unfortunately, the GBE has only been cloned together with a SpCas9 [9], for which there is not a PAM site that would put the target into the editing window. However, such a construct, utilising a different PAM site that is available in the target area, could be developed by cloning.

Alternatively, the mutated amino acid, cysteine, could be edited to another, better tolerated amino acid. We have described all the editing options in Table 1, with options 2–5 describing this alternative approach. Interestingly, editing the cysteine to arginine (option 3) was predicted in silico to be non-pathogenic by SIFT, PolyPhen-2 and Mutation Taster and there is an available NG-SpCas9-ABE PAM site that enabled us to design a guideRNA, ATGATGC[A>T]TTTATCCGGGGGgg (mutation in brackets, PAM site in lower case letters), targeting the reverse allele. There does, however, appear to be one likely bystander edit in this approach, p.(Ile205Thr). This bystander was found to result in a benign variant by Mutation Taster (Polymorphism; Model: *simple_aae*, prob: 0.920824664343818), Polyphen2 (benign, score of 0.000 (sensitivity: 1.00; specificity: 0.00) and SIFT (tolerated, score = 0.28).

Editing the cysteine into tyrosine (option 5) is predicted to affect protein function according to SIFT and Mutation Taster but is benign based on PolyPhen-2 analysis. There is low confidence in this prediction (Table 2). The conflicting predictions might result from the fact that the correlation of computational algorithms with functional studies may not be as consistent in proteins that do not have high phylogenetic conservation rates. SIFT software is based on the assumption that evolutionarily conserved regions are more likely to be less tolerant of variants, and hence alterations in these regions are more likely to adversely affect protein function [20]. SIFT utilises sequence homology to calculate the probability that a certain amino acid change will affect protein function [20].

## 4. Discussion

In this report, we describe the clinical findings and genetic testing results from an SFD patient. This report highlights the importance of genetic testing in young patients with CNV, particularly within the context of a strong family history of presumed wet age-related macular degeneration. We also propose CRISPR DNA base editing options as a plausible method to develop new therapies to treat TIMP3-associated SFD.

The clinical presentation of this patient, with evidence of bilateral CNV, would be consistent with a diagnosis of early-onset neovascular disease. A list of differential diagnoses includes SFD, pattern dystrophy (associated gene *PRPH2*), or other dominantly inherited macular dystrophies, including late stage Best disease (*BEST1*), and central serous retinopathy with secondary CNV [24].

An early diagnosis of SFD is especially important for prompt institution of anti-VEGF injections to preserve central vision, monitoring of the disease, and for appropriate counselling. CNV in patients with SFD are highly responsive to anti-vascular endothelial growth factor antibodies with treatment effectively delaying visual loss and scarring [3,25,26].

A diagnosis of SFD is also important as a retrospective, observational case series has shown that some patients initially diagnosed with neovascular AMD carry the SFD-associated TIMP3 mutation p.(Ser38Cys) [27]. This missense mutation has been identified by the International AMD Genomics Consortium (IAMDGC); hence, care should be taken when considering early AMD cohorts as they may contain some SFD patients. Identifying such patients may contribute to the identification of novel TIMP3 variants. Molecular analysis and studies of their downstream functional consequences may help elucidate the shared pathophysiological mechanisms in these diseases, potentially leading to the development of novel therapies for diseases in which TIMP3 accumulation has also been observed—these include more severe forms of AMD, and through indirect mechanisms, Alzheimer’s disease [28,29].

An additional finding in our patient included a novel mutation in the *ABCA4* gene. The *ABCA4* c.4601del, p.(Leu1534fs) is predicted to lead to premature termination of translation. As far as we are aware, this variant has not been described in the literature; however, other *ABCA4* variants predicted to cause premature termination of translation have been detected in individuals with macular dystrophy [30,31].The consequences of this variant cannot be confirmed without supporting in vitro data; however, we can hypothesise that harbouring this additional variant may explain the intrafamilial phenotypic variability and possible earlier onset of pathology in this patient, especially as the VFVNFA motif in the ABCA4 amino acid sequence is phylogenetically highly conserved [32]. There is no prior evidence in the literature, to our knowledge, of simultaneous mutations in the TIMP3 and *ABCA4* genes.

There are currently no standardised, curative treatment options for TIMP3-associated SFD. While gene replacement therapy has proven its safety and efficacy in treating recessive *RPE65*-associated Leber congenital amaurosis [9], such an approach is not suitable for dominant-negative diseases due to the need to silence the defective allele. A new tempting approach are the CRISPR DNA base editors that are able to correct all transition an even some transversion variants [10,11,12]. While the TIMP3 c.610A>T, p.(Ser204Cys) cannot be edited back to wild-type on DNA-level, we propose in Table 2 multiple options to either correct the variant on the amino acid level, or to edit the mutated residue into a better tolerated one. Extensive in vitro and in vivo studies are needed to show safety and efficacy of these approaches, but they might offer plausible new ways to develop treatments for SFD patients. Recently, Choi et al. have shown that in vivo correction of an *Rpe65* mutation by subretinal injection of adenine base editor (ABE) and sgRNA, prolonged the survival of cones in a Leber congenital amaurosis mouse model [33]. Weber et al. have generated a knock-in mouse model for Sorsby which carries the disease-related Ser156Cys mutation, and this model could be used to study in vivo safety and efficacy of new therapeutic options [34]. Based on our analyses, construct-specific PAM site requirements are a limiting factor for base editing options; however, this difficulty will likely be eased in the future with new nearly PAM-less constructs being developed [35].

Serotypes of adeno-associated viruses (AAV) used in clinical trials of inherited retinal disease (IRD) thus far include AAV2/5, AAV2/8 and AAV8 [36]. To bypass the limited cargo capacity of AAVs, 4.7 kB, dual AAV vectors have been used in murine models of IRDs. The reconstitution of the full-length expression cassette is accomplished upon co-infection of both dual AAV vectors in the same cell. The eye tends to favour such co-infection due to the discrete subretinal space. Trapani et al. demonstrated that dual AAV vectors may have a reduced photoreceptor transduction efficiency—observed to be 6% and 40% of that of a single AAV vector in mice and pigs, respectively [37]. It is therefore postulated that very high AAV doses are required which may increase the risk of gene therapy -related uveitis.

Frequent clinic visits for anti-VEGF treatment are costly and time-intensive. A one-time gene therapy before the onset of choroidal neovascularisation and fibrosis could reduce this treatment burden. With respect to inflammation, the SFD patient population is older than the IRD population receiving gene therapy (developing CNV at about 40 years of age). This may be advantageous as immune reactions are less likely to be as exuberant.

Instead of correcting the pathogenic variants, base editors could also be utilised to introduce mutant-specific stop codons to silence the defective allele. However, TIMP3 knockout mice have been found to develop irregular choroidal vessels with dilated capillaries and showed increased MMP activity in the choroid [38]. A null mutation may also be of concern as TIMP3 knock out mice have present with pulmonary alveolar enlargement [39], enhanced susceptibility to cardiomyopathy [40], and hepatic injury [41], highlighting the need to silence selectively only the mutated allele.

As a conclusion, we report here the clinical and genetic findings from an SFD patient with a heterozygous TIMP3 c.610A>T, p.Ser204Cys variant. Base editing might offer new treatment options for autosomal dominant diseases such as SFD, but further studies are needed prior to their use in patients. 

## Figures and Tables

**Figure 1 genes-13-02103-f001:**
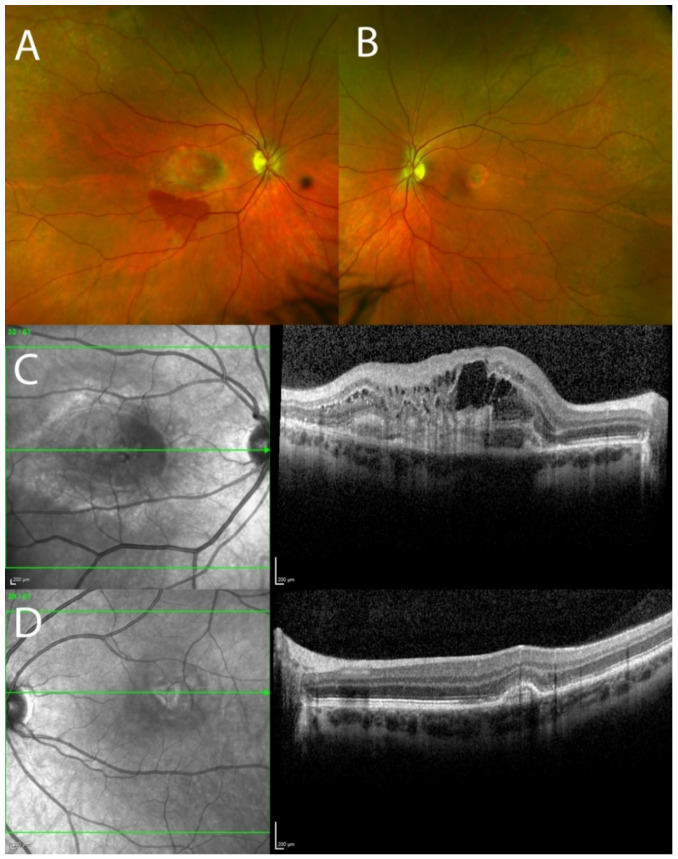
Pseudo-colour fundus Optos images and optical coherence tomography (OCT) showing choroidal neovascularisation (CNV) in the macula in both eyes. (**A**) Right fundus image showing greyish lesion at the fovea and intraretinal haemorrhage below the lesion; (**B**) Left fundus image showing juxtafoveal greyish lesion; (**C**) OCT of the macula of the right eye showing intraretinal cysts and extensive subretinal hyperreflective material; (**D**) OCT of the macula of the left eye showing fibrovascular pigment epithelial detachment.

**Figure 2 genes-13-02103-f002:**
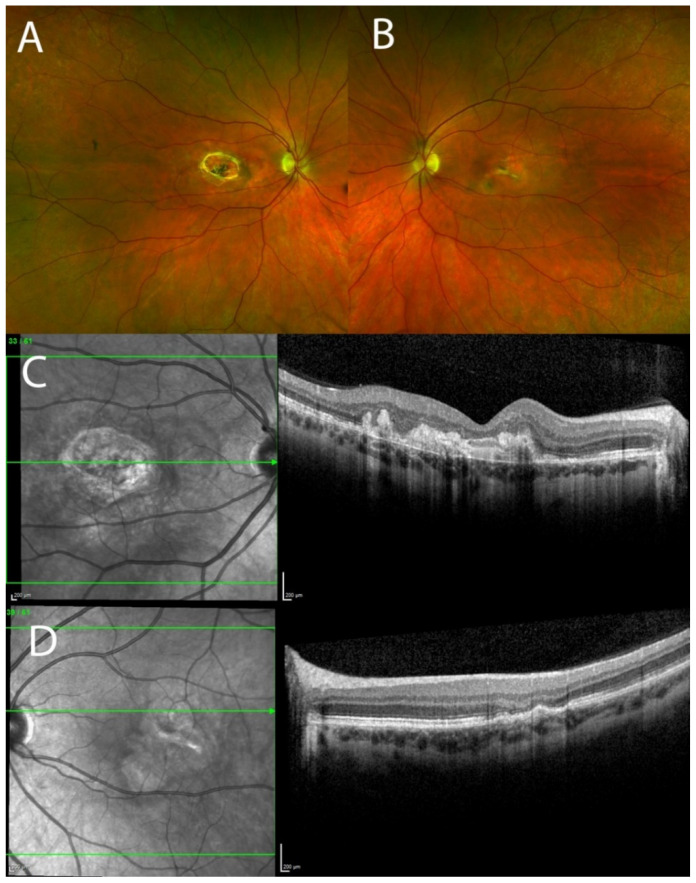
Pseudo-colour fundus Optos images and optical coherence tomography (OCT) showing disciform scar in the right eye and inactive choroidal neovascularisation (CNV) in the left eye 31 months after the treatment with aflibercept. (**A**,**C**) Inactive disciform scar in the right eye; (**B**,**D**) No neovascular activity with juxtafoveal scarring and retinal atrophy.

**Table 1 genes-13-02103-t001:** Base editor constructs included in the analyses.

Construct	Ideal Editing Window *	PAM Site *	Reference
SpCas9-ABE8e	4′–8′	20′ NGG	[14]
SaCas9-ABE8e	3′–14′	21′ NNGRRT	[14]
KKH-SaCas9-ABE8e	3′–14′	21′ NNNRRT	[14]
CasMINI-ABE8e	3′–4′	Upstream of the target base: TTTR	[15]
NG-SpCas9-ABE	4′–8′	20′NG	[16]
SpCas9-CBE	4′–8′	20′ NGG	[17]
SaCas9-CBE	2′–12′	21′ NNGRRT	[17]
SpCas9-GBE	6′	20′ NGG	[18]

N = A, T, G or C; R = A or T; * Numbers refer to nucleotide positions in the guide-RNA.

**Table 2 genes-13-02103-t002:** Base editing options to treat the dominant TIMP3 c.610A>T, p.(Ser204Cys).

Option	Base Editor	Strand	Codon Change Patient > Edited	Amino Acid Change	SIFT	PolyPhen-2 Score	Mutation Taster Score
1	GBE	Reverse	TGC > TCC	p.Ser204Ser	Wild-type amino acid	Wild-type amino acid	Wild-type amino acid
2	GBE	Forward	TGC > TGG	p.Ser204Trp	Affects protein function, score = 0.01. Median sequence conservation: 3.37.	Possibly damaging with a score of 0.721 (sensitivity: 0.86; specificity: 0.92)	Polymorphism (Model: without_aae, prob: 0.999999181161545). Protein features (might be) affected—splice site changes.
3	ABE	Reverse	TGC > CGC	p.Ser204Arg	Tolerated, score = 0.06. Median sequence conservation: 3.37.	Benign, score = 0.023. Sensitivity: 0.95; specificity: 0.81.	Polymorphism (Model: without aae, prob: 0.999999621231165).
4	CBE	Forward	TGC > TGT	p.Ser204Cys	Identical to patient mutation	Identical to patient mutation	Identical to patient mutation
5	CBE	Reverse	TGC > TAC	p.Ser204Tyr	Affects protein function, score = 0.02. Median sequence conservation: 3.37.	Benign, score = 0.183. Sensitivity: 0.92; specificity: 0.87.	Polymorphism (Model: without_aae, prob: 0.999999618732328). Protein features (might be) affected—splice site changes

## Data Availability

Not applicable.

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
