# Peer review of "Potential CRISPR Base Editing Therapeutic Options in a Sorsby Fundus Dystrophy Patient"

_genes, 2022, doi:10.3390/genes13112103_

Round 1

Reviewer 1 Report

The authors reported a patient suffering from Sorsby fundus dystrophy. The patient was phenotyped and genotyped. Potential causative variants were discovered in the TIMP3 gene. Possible therapeutic options were explored for this specific patient using base editing techniques. Describing these clinical cases is important since knowing the causative variants is important when deciding on the best therapeutic strategy in many cases. 

General

While reading the title, I was interested in the details regarding the therapeutic options of the paper. To be honest, I expected at least some in vitro data while reading the title. The authors could consider clarifying in the title that the article contains bioinformatic analysis data regarding the therapeutic options. Is classification as an Article correct, or is this more a (case) report?

Comments, questions and suggestions:

1.       While reading the the introduction I was wondering about the TIMP3  gene, the structure of the protein and its function. Later, in the discussion section,  I found more information. Maybe some of that information could be moved to the introduction.

2.       Section 2.1: Could the authors add the timing of the clinical evaluations and the treatments given to the patient? At what age did the patient have the first symptoms?

3.       Discussion section: “Extensive in vitro and in vivo studies are needed to show……”. Could the authors give some examples of the studies they would / are planning to perform?

Author Response

Review comments 1:

Title of the manuscript: Potential base editing therapeutic options in a patient presenting with Sorsby fundus dystrophy

The authors reported a patient suffering from Sorsby fundus dystrophy. The patient was phenotyped and genotyped. Potential causative variants were discovered in the TIMP3 gene. Possible therapeutic options were explored for this specific patient using base editing techniques. Describing these clinical cases is important since knowing the causative variants is important when deciding on the best therapeutic strategy in many cases.

Authors' response: We thank the reviewer for taking the time to read our manuscript, and for providing helpful comments to improve it even further.

General

While reading the title, I was interested in the details regarding the therapeutic options of the paper. To be honest, I expected at least some in vitro data while reading the title. The authors could consider clarifying in the title that the article contains bioinformatic analysis data regarding the therapeutic options. Is classification as an Article correct, or is this more a (case) report?

Authors' response: We thank the reviewer for this suggestion. We have changed the title to: Potential CRISPR base editing therapeutic options in a patient presenting with Sorsby fundus dystrophy. To avoid a lengthy title, we have added the following sentence to the abstract:

Line 14 and 15:We  report a bioinformatic analysis of potential therapeutic options in a patient presenting with Sorsby fundus dystrophy.

  1. While reading the the introduction I was wondering about the TIMP3gene, the structure of the protein and its function. Later, in the discussion section,  I found more information. Maybe some of that information could be moved to the introduction.

Authors' response: We thank the reviewer for this suggestion. We have added the information about the structure of the protein and function in the introduction instead of discussion.

  1. Section 2.1: Could the authors add the timing of the clinical evaluations and the treatments given to the patient? At what age did the patient have the first symptoms?

Authors' response: We thank the reviewer for this suggestion. The patient had his first symptoms at the age of 35 (Line 127). We have added more information about the course of the treatment/visits, please see lines 150-154:

In the first 3 years of treatment, the patient received 22 intravitreal injections of aflibercept in the right eye, and 20 injections in the left eye. Due to the waxing and waning of neovascular activity in both eyes, the patient was either on one-monthly or two-monthly intervals between the injections or on a pro re nata protocol.

  1. Discussion section: “Extensive in vitroand in vivo studies are needed to show……”. Could the authors give some examples of the studies they would / are planning to perform?

We thank the reviewer for this suggestion. We have added more information in response to this point. Please see lines 286-291:

Recently, Choi et al have shown that in vivo correction of an Rpe65 mutation by subretinal injection of adenine base editor (ABE) and sgRNA, prolonged the survival of cones in a Leber congenital amaurosis mouse model. Weber et al have generated a knock-in mouse model for Sorsby which carries the disease-related Ser156Cys mutation, and this model could be used to study in vivo safety and efficacy of new therapeutic options.

Reviewer 2 Report

In this study Elsayed et al. present a case study of a 35-year-old patient with Sorsby fundus dystrophy (SFD) and propose some therapeutic options using Base editing for such cases. While, the options presented are interesting I have a general question about their proposed strategy that is not clear from the manuscript – 

1.     The authors propose that the Cysteine at 204 can be converted to better tolerated amino acids and these ‘better tolerated’ aa are based on predictions by the SIFT software. But, is there any evidence (from human samples) that these alternatives will indeed be safe? Probably a brief explanation/ comment on how the SIFT software makes these predictions can suffice. 

In the discussion the authors need to mention some aspects about how feasible these options will be-

1.     For example, presently all ocular gene therapies are provided by packaging the gene into AAVs and delivering by intravitreal/subretinal injections. Since the Base editors are too large to fit into a single AAV they need to be split to fit into two or a new generation of shorter versions of BEs need to be used. The authors need to comment on this in the discussion.

2.     If dual AAVs are used additional comment is required on the level of transduction achieved and hence the therapeutic benefits possible. Or discuss alternative methods for delivery of BE to the eye.

3.     The authors present a case of a patient with quite advanced SFD, so in the discussion it is worth mentioning the age at which the proposed therapies should be provided for maximum benefit and also comment on whether it is worth treating advanced patients to prevent further degeneration?

Minor corrections-

Figure 2 – Legend does not describe panel D

Discussion – Lines 188-192 repeated

Author Response

Review comments 2:

Title of the manuscript: Potential base editing therapeutic options in a patient presenting with Sorsby fundus dystrophy.

In this study Elsayed et al. present a case study of a 35-year-old patient with Sorsby fundus dystrophy (SFD) and propose some therapeutic options using Base editing for such cases. While, the options presented are interesting I have a general question about their proposed strategy that is not clear from the manuscript – 

  1. The authors propose that the Cysteine at 204 can be converted to better tolerated amino acids and these ‘better tolerated’ aa are based on predictions by the SIFT software. But, is there any evidence (from human samples) that these alternatives will indeed be safe? Probably a brief explanation/ comment on how the SIFT software makes these predictions can suffice. 

We thank the reviewer for their suggestion and have added such comment. Please see lines 206-210:

SIFT software is based on the assumption that evolutionarily conserved regions are more likely to be less tolerant of variants, and hence alterations in these regions are more likely to adversely affect protein function. SIFT utilizes sequence homology to calculate the probability that a certain amino acid change will affect protein function.

In the discussion the authors need to mention some aspects about how feasible these options will be-

  1. For example, presently all ocular gene therapies are provided by packaging the gene into AAVs and delivering by intravitreal/subretinal injections. Since the Base editors are too large to fit into a single AAV they need to be split to fit into two or a new generation of shorter versions of BEs need to be used. The authors need to comment on this in the discussion.

We thank the reviewer for this suggestion. Please see lines 293-302:

Serotypes of adeno-associated viruses (AAV) used in clinical trials of inherited retinal disease (IRD) thus far include AAV2/5, AAV2/8 and AAV8. To bypass the limited cargo capacity of AAVs, 4.7kB, dual AAV vectors have been used in murine models of IRDs. The reconstitution of the full-length expression cassette is accomplished upon co-infection of both dual AAV vectors in the same cell. The eye tends to favour such co-infection due to the discrete subretinal space. Trapani et al demonstrated that dual AAV vectors may have a reduced photoreceptor transduction efficiency - observed to be 6% and 40% of that of a single AAV vector in mice and pigs, respectively. It is therefore postulated that very high AAV doses are required which may increase the risk of gene therapy -related uveitis.

  1. If dual AAVs are used additional comment is required on the level of transduction achieved and hence the therapeutic benefits possible. Or discuss alternative methods for delivery of BE to the eye.

We thank the reviewer for this suggestion. Please see lines 299-303:

Trapani et al demonstrated that dual AAV vectors may have a reduced photoreceptor transduction efficiency - observed to be 6% and 40% of that of a single AAV vector in mice and pigs, respectively. It is therefore postulated that very high AAV doses are required which may increase the risk of gene therapy -related uveitis.

.

  1. The authors present a case of a patient with quite advanced SFD, so in the discussion it is worth mentioning the age at which the proposed therapies should be provided for maximum benefit and also comment on whether it is worth treating advanced patients to prevent further degeneration?

We thank the reviewer for this suggestion. Please see lines 303-307:

Frequent clinic visits for anti-VEGF treatment is costly and time-intensive. A one-time gene therapy before the onset of choroidal neovascularisation and fibrosis could reduce this treatment burden. With respect to inflammation, the SFD patient population is older than the IRD population receiving gene therapy (developing CNV at about 40 years of age). This may be advantageous as immune reactions are less likely to be as exuberant (REF)

Minor corrections-

Figure 2 – Legend does not describe panel D

Authors' response: Thank you, this has been corrected in the revised document.

Discussion – Lines 188-192 repeated

Authors' response: Thank you, this has been corrected in the revised document.